# Understanding Interlocking Dynamics of Cooperative Rationalization

**Mo Yu**[1*]  **Yang Zhang**[1*]  **Shiyu Chang**[1,2*]  **Tommi S. Jaakkola**[3]
[1]MIT-IBM Watson AI Lab    [2]UC Santa Barbara    [3]CSAIL MIT
yum@us.ibm.com  yang.zhang2@ibm.com  chang87@ucsb.edu
tommi@csail.mit.edu

## Abstract

Selective rationalization explains the prediction of complex neural networks by finding a small subset of the input that is sufficient to predict the neural model output. The selection mechanism is commonly integrated into the model itself by specifying a two-component cascaded system consisting of a rationale generator, which makes a binary selection of the input features (which is the rationale), and a predictor, which predicts the output based only on the selected features. The components are trained jointly to optimize prediction performance. In this paper, we reveal a major problem with such cooperative rationalization paradigm — *model interlocking*. Interlocking arises when the predictor overfits to the features selected by the generator thus reinforcing the generator's selection even if the selected rationales are sub-optimal. The fundamental cause of the interlocking problem is that the rationalization objective to be minimized is concave with respect to the generator's selection policy. We propose a new rationalization framework, called A2ʀ, which introduces a third component into the architecture, a predictor driven by soft attention as opposed to selection. The generator now realizes both soft and hard attention over the features and these are fed into the two different predictors. While the generator still seeks to support the original predictor performance, it also minimizes a gap between the two predictors. As we will show theoretically, since the attention-based predictor exhibits a better convexity property, A2ʀ can overcome the concavity barrier. Our experiments on two synthetic benchmarks and two real datasets demonstrate that A2ʀ can significantly alleviate the interlock problem and find explanations that better align with human judgments.[2]

## 1 Introduction

Selective rationalization [8, 10, 11, 13, 14, 17, 27, 29, 46] explains the prediction of complex neural networks by finding a small subset of the input – rationale – that suffices on its own to yield the same outcome as to the original data. To generate high-quality rationales, existing methods often train a cascaded system that consists of two components, *i.e.*, a *rationale generator* and a *predictor*. The generator selects a subset of the input explicitly (*a.k.a.*, binarized selection), which is then fed to the predictor. The predictor then predicts the output based only on the subset of features selected by the generator. The rationale generator and the predictor are trained jointly to optimize the prediction performance. Compared to many other interpretable methods [5, 23, 45, 38] that rely on attention mechanism as a proxy of models' explanation, selective rationalization offers a unique advantage: certification of exclusion, *i.e.*, any unselected input is guaranteed to have no contribution to prediction.

---

[*]Authors contributed equally to this paper. Work was done when SC was at MIT-IBM Watson AI Lab.

[2]We release our code at https://github.com/Gorov/Understanding_Interlocking.

35th Conference on Neural Information Processing Systems (NeurIPS 2021).

However, binarized selective rationalization schemes are notoriously hard to train [8, 46]. To overcome training obstacles, previous works have considered using smoothed gradient estimations (*e.g.* gradient straight-through [9] or Gumbel softmax [21]), introducing additional components to control the complement of the selection [10, 46], adopting different updating dynamics between the generator and the predictor [11], using rectified continuous random variables to handle the constrained optimization in training [8], *etc*. In practice, these solutions are still insufficient. They either still require careful tuning or are at a cost of reduced predictive accuracy.

In this paper, we reveal a major training problem of selective rationalization that has been largely overlooked — *model interlocking*. Intuitively, this problem arises because the predictor only sees what the generator selects during training, and tends to overfit to the selection of the generator. As a result, even if the generator selects a sub-optimal rationale, the predictor can still produce a lower prediction loss when given this sub-optimal rationale than when given the optimal rationale that it has never seen. As a result, the generator's selection of the sub-optimal rationale will be reinforced. In the end, both the rationale generator and the predictor will be trapped in a sub-optimal equilibrium, which hurts both model's predictive accuracy and the quality of generated rationales.

By investigating the training objective of selective rationalization theoretically, we found that the fundamental cause of the problem of interlocking is that the rationalization objective we aim to minimize is undesirably *concave* with respect to the rationale generator's policy, which leads to many sub-optimal corner solutions. On the other hand, although the attention-based models (*i.e.*, via soft selection) produce much less faithful explanations and do not have the nice property of certification of exclusion, their optimization objective has a better convexity property with respect to the attention weights under certain assumptions, and thus would not suffer from the interlocking problem.

Motivated by these observations, we propose a new rationalization framework, called A2R (attention-to-rationale), which combines the advantages of both the attention model (convexity) and binarized rationalization (faithfulness) into one. Specifically, our model consists of a generator, and two predictors. One predictor, called *attention-based predictor*, operates on the soft-attention, and the other predictor, called *binarized predictor*, operates on the binarized rationales. The attention as used by the attention-based predictor is tied to the rationale selection probability as used by the binarized predictor. During training, the generator aims to improve both predictors' performance while minimizing their prediction gap. As we will show theoretically, the proposed rationalization scheme can overcome the concavity of the original setup, and thus can avoid being trapped in sub-optimal rationales. In addition, during inference time, we only keep the binarized predictor to ensure the faithfulness of the generated explanations. We conduct experiments on two synthetic benchmarks and two real datasets. The results demonstrate that our model can significantly alleviate the problem of interlocking and find explanations that better align with human judgments.

## 2 Related Work

**Selective rationalization:** [27] proposes the first generator-predictor framework for rationalization. Following this work, new game-theoretic frameworks were proposed to encourage different desired properties of the selected rationales, such as optimized Shapley structure scores [14], comprehensiveness [46], multi-aspect supports [4, 11] and invariance [12]. Another fundamental direction is to overcome the training difficulties. [6] replaces policy gradient with Gumbel softmax. [46] proposes to first pre-train the predictor, and then perform end-to-end training. [11] adopts different updating dynamics between the generator and the predictor. [8] replaces the Bernoulli sampling distributions with rectified continuous random variables to facilitate constrained optimization. [39] proposes to enhance the training objective with an adversarial information calibration according to a black-box predictor. However, these methods cannot address the problem of interlocking.

**Attention as a proxy of explanation:** Model's attention [5, 23, 45] could serve as a proxy of the rationale. Although attention is easy to obtain, it lacks faithfulness. An input associated with low attention weight can still significantly impact the prediction. In addition, recent works [7, 20, 35, 38, 44] also find that the same prediction on an input could be generated by totally different attentions, which limits its applicability to explaining neural predictions. To improve the faithfulness of attentions, [33, 43] regularize the hidden representations on which the attention is computed over; [17] applies attention weights on losses of pre-defined individual rationale candidates' predictions. Nevertheless, rationales remain to be more faithful explanations due to their certification of exclusion.

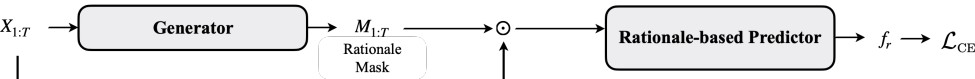

Figure 1: A conventional selective rationalization framework.

[18, 42] force the sparsity of the attention with sparsemax [31], so as to promote the faithfulness of their attention as rationales. The interlocking problem still persists in this framework, because the loss landscape remains concave (refer to our arguments in Section 3.2&3.3). Specifically, since the predictor would not see the sentences that receive 0 attention weights, it tends to underfit these sentences. As a result, the generator does not have the incentive to assign positive weights to the sentences that are previously assigned zero weights, thus is prone to selecting the same sentences.

**Model interpretability beyond selective rationalization:** There are other popular interpretability frameworks besides selective rationalization. Module networks [2, 3, 22] compose appropriate neural modules following a logical program to complete the task. Their applicability is relatively limited, due to the requirement of pre-defined modules and programs. Evaluating feature importance with gradient information [7, 28, 40, 41] is another popular method. Though [7] discusses several advantages of gradient-based methods over rationalization, they are post-hoc and cannot impose structural constraints on the explanation. Other lines of work that provide post-hoc explanations include local perturbations [25, 30]; locally fitting interpretable models [1, 36]; and generating explanations in the form of edits to inputs that change model prediction to the contrast case [37].

## 3 Selective Rationalization and Interlocking

In this section, we will formally analyze the problem of interlocking in conventional selective rationalization frameworks. Throughout this section, upper-cased letters, *i.e.*, $\boldsymbol{A}$ and $A$, represent random vectors (bolded) and random values (unbolded) respectively; lower cased letters, *i.e.*, $\boldsymbol{a}$ and $a$, represent deterministic vectors (bolded) and values (unbolded) respectively. Vectors with a colon subscript, *i.e.*, $\boldsymbol{a}_{1:T}$, represent a concatenation of $\boldsymbol{a}_1$ to $\boldsymbol{a}_T$, *i.e.*, $[\boldsymbol{a}_1; \cdots ; \boldsymbol{a}_T]$.

### 3.1 Overview of Selective Rationalization

Consider a classification problem, $(\boldsymbol{X}, Y)$, where $\boldsymbol{X} = \boldsymbol{X}_{1:T}$ is the input feature, and $Y$ is the discrete class label. In NLP applications, $\boldsymbol{X}_{1:T}$ can be understood as a series of $T$ words/sentences. The goal of selective rationalization is to identify a *binary* mask, $\boldsymbol{M} \in \{0, 1\}^T$, that applies to the input features to form a rationale vector, $\boldsymbol{Z}$, as an explanation of $Y$. Formally, the rationale vector $\boldsymbol{Z}$ is defined as

$$\boldsymbol{Z} = \boldsymbol{M} \circ \boldsymbol{X} \equiv [M_1 \boldsymbol{X}_1, \cdots, M_T \boldsymbol{X}_T]. \tag{1}$$

Conventionally, $\boldsymbol{Z}$ is determined by maximizing the mutual information between $\boldsymbol{Z}$ and $Y$, *i.e.*,

$$\max_{\boldsymbol{M}} I(Y; \boldsymbol{M} \circ \boldsymbol{X}), \quad \text{s.t. } \boldsymbol{M} \in \mathcal{M}, \tag{2}$$

where $\mathcal{M}$ refers to a constraint set, such as the sparsity constraint and a continuity constraint, requiring that the selected rationale should be a small and continuous subset of the input features.

One way of learning to extract the rationale under this criterion is to introduce a game-theoretic framework (see Figure 1) consisting of two players, a rationale generator and a predictor. The rationale generator selects a subset of input as rationales and the predictor makes the prediction based only on the rationales. The two players cooperate to maximize the prediction accuracy, so the rationale generator would need to select the most informative input subset.

Specifically, the rationale generator generates a probability distribution, $\boldsymbol{\pi}$, for the masks, based on the input features $\boldsymbol{X}$. Then, the mask $\boldsymbol{M}$ is randomly drawn from the distribution $\boldsymbol{\pi}$. To simplify our exposition, we focus on the case that $\boldsymbol{X}_i$ represents a sentence and only one of the $T$ sentences is selected as a rationale. In this case, $\boldsymbol{M}$ is a one-hot vector, and $\boldsymbol{\pi}$ is a multinomial distribution. Formally, the mask $\boldsymbol{M}$ is generated as follows

$$\boldsymbol{M} \sim \boldsymbol{\pi}(\boldsymbol{X}) = [\pi_1(\boldsymbol{X}), \cdots, \pi_T(\boldsymbol{X})], \quad \text{where } \pi_i(\boldsymbol{X}) = p(\boldsymbol{M} = \boldsymbol{e}_i | \boldsymbol{X})],$$

and $\boldsymbol{e}_i$ denotes a $T$-dimensional one-hot vector, with the $i$-th dimension equal to one. The generalization to making multiple selections will be discussed in Section 4.1.

After the mask is generated, the predictor, $\boldsymbol{f}_r(\cdot; \boldsymbol{\theta}_r)$ (the subscript $r$ stands for rationale to differentiate from the attention-based predictor introduced later), then predicts the probability distribution of $Y$ based only on $\boldsymbol{Z} = \boldsymbol{M} \circ \boldsymbol{X}$, *i.e.*,

$$\boldsymbol{f}_r(\boldsymbol{Z}; \boldsymbol{\theta}_r) = [\hat{p}(Y = 1|\boldsymbol{Z}), \cdots, \hat{p}(Y = c|\boldsymbol{Z})], \tag{3}$$

where $\hat{p}$ represents a predicted distribution, and $\boldsymbol{\theta}_r$ denotes the parameters of the predictor.

The generator and the predictor are trained jointly to minimize the cross-entropy loss of the prediction:

$$\min_{\boldsymbol{\pi}(\cdot), \boldsymbol{\theta}_r} \mathcal{L}_r(\boldsymbol{\pi}, \boldsymbol{\theta}_r), \quad \text{where } \mathcal{L}_r(\boldsymbol{\pi}, \boldsymbol{\theta}_r) = \mathbb{E}_{\substack{\boldsymbol{X},Y \sim \mathcal{D}_{tr} \\ \boldsymbol{M} \sim \boldsymbol{\pi}(\boldsymbol{X})}}[\ell(Y, \boldsymbol{f}_r(\boldsymbol{M} \circ \boldsymbol{X}; \boldsymbol{\theta}_r))]. \tag{4}$$

$\mathcal{D}_{tr}$ denotes the training set; $\ell(\cdot, \cdot)$ denotes the cross entropy loss. It can be shown [13] that, if $\boldsymbol{\pi}(\cdot)$ and $\boldsymbol{f}_r(\cdot; \boldsymbol{\theta}_r)$ both have sufficient representation power, the globally optimal $\boldsymbol{\pi}(\boldsymbol{X})$ of Equation (4) would generate masks $\boldsymbol{M}$ that are globally optimal under Equation (2).

## 3.2 Interlocking: A Toy Example

Despite the nice guarantee of its global optimum solution, the rationalization framework in Equation (4) suffers from the problem of being easily trapped into poor local minima, a problem we refer to as *interlocking*. To help readers understand the nature of this problem, we would like to start with a toy example, where the input consists of two sentences, $\boldsymbol{X}_1$ and $\boldsymbol{X}_2$. We assume that $\boldsymbol{X}_1$ is the more informative (in terms of predicting $Y$) sentence between the two, so the optimal solution for the rationale generator $\boldsymbol{\pi}$ is to always select $\boldsymbol{X}_1$ (*i.e.* $\pi_1 = 1$, and $\pi_2 = 0$).

However, assume, for some reason, that the generator is initialized so poorly that it only selects $\boldsymbol{X}_2$, and that the predictor has been trained to make the prediction based only on $\boldsymbol{X}_2$. In this case, we will show that it is very hard for the generator-predictor to escape from this poor local minimum thus it fails to converge to the globally optimal solution of selecting $\boldsymbol{X}_1$. Since the predictor underfits to $\boldsymbol{X}_1$, it will produce a large prediction error when $\boldsymbol{X}_1$ is fed. As a result, the rationale generator would stick with selecting $\boldsymbol{X}_2$ because $\boldsymbol{X}_2$ yields a smaller prediction error than $\boldsymbol{X}_1$. The predictor, in turn, would keep overfitting to $\boldsymbol{X}_2$ and underfitting to $\boldsymbol{X}_1$. In short, both players lock the other player from escaping from the poor solution, hence the name interlocking.

The problem of interlocking can also be manifested by an accordance game, where the generator has two strategies, *select $\boldsymbol{X}_1$* and *select $\boldsymbol{X}_2$*, and the predictor also has two strategies, *overfit to $\boldsymbol{X}_1$* and *overfit to $\boldsymbol{X}_2$*. An example payoff table is shown in Table 1. As can be seen, (*select $\boldsymbol{X}_1$, overfit to $\boldsymbol{X}_1$*) has the highest payoff, and thus is the optimal solution for both players. However, (*select $\boldsymbol{X}_2$, overfit to $\boldsymbol{X}_2$*) also constitutes a Nash equilibrium, which is locally optimal.

Table 1: An example payoff (negative loss) table of the accordance game between the generator (Gen) and the predictor (Pred), where the interlocking problem is manifested as multiple Nash Equilibria.

|      |                | Pred. | |
|------|----------------|----------------------|----------------------|
|      |                | *Overfit to $\boldsymbol{X}_1$* | *Overfit to $\boldsymbol{X}_2$* |
| **Gen.** | *Select $\boldsymbol{X}_1$* | $(-1, -1)$ | $(-10, -10)$ |
|      | *Select $\boldsymbol{X}_2$* | $(-20, -20)$ | $(-2, -2)$ |

## 3.3 Interlocking and Concave Minimization

To understand the fundamental cause of the interlocking problem, rewrite the optimization problem in Equation (4) into a nested form:

$$\min_{\boldsymbol{\pi}(\cdot), \boldsymbol{\theta}_r} \mathcal{L}_r(\boldsymbol{\pi}, \boldsymbol{\theta}_r) = \min_{\boldsymbol{\pi}(\cdot)} \min_{\boldsymbol{\theta}_r} \mathcal{L}_r(\boldsymbol{\pi}, \boldsymbol{\theta}_r) = \min_{\boldsymbol{\pi}(\cdot)} \mathcal{L}_r(\boldsymbol{\pi}, \boldsymbol{\theta}_r^*(\boldsymbol{\pi})), \tag{5}$$

$$\text{where} \quad \boldsymbol{\theta}_r^*(\boldsymbol{\pi}) = \operatorname*{argmin}_{\boldsymbol{\theta}_r} \mathcal{L}_r(\boldsymbol{\pi}, \boldsymbol{\theta}_r). \tag{6}$$

Furthermore, denote

$$\mathcal{L}_r^*(\boldsymbol{\pi}) = \mathcal{L}_r(\boldsymbol{\pi}, \boldsymbol{\theta}_r^*(\boldsymbol{\pi})). \tag{7}$$

Then, the problem of finding the optimal rationale boils down to finding the global minimum of $\mathcal{L}_r^*(\boldsymbol{\pi})$. In order to achieve good convergence properties, $\mathcal{L}_r^*(\boldsymbol{\pi})$ would ideally be convex with respect to $\boldsymbol{\pi}$. However, the following theorem states the opposite.

**Theorem 1.** $\mathcal{L}_r^*(\boldsymbol{\pi})$ *is concave with respect to* $\boldsymbol{\pi}$.

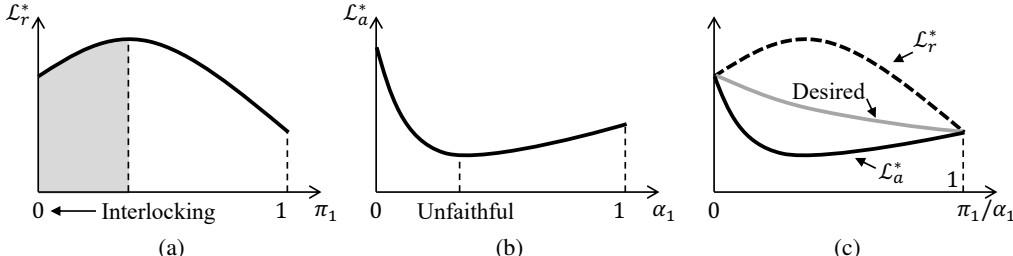

Figure 2: Example loss landscapes of the two-sentence scenario. (a) An example loss landscape of rationale-based explanation (Equation (7)), which is concave and induces interlocking dynamics towards a sub-optimal local minimum. (b) An example loss landscape of attention-based explanation (Equation (9)), which is convex but with an unfaithful global minimum. (c) The two loss landscapes share common end points. Desirable landscapes should lie in between.

The proof is presented in Appendix A.1. Theorem 1 implies the cooperative rationalization objective can contain many local optima at the corners. Going back to the two-sentence example, Figure 2(a) plots an example $\mathcal{L}_r^*(\boldsymbol{\pi})$ against $\pi_1$. Since there are two sentences, $\pi_1 = 0$ implies that the generator always selects $\boldsymbol{X}_2$, and $\pi_1 = 1$ implies the generator always selects $\boldsymbol{X}_1$. As shown in the figure, since $\boldsymbol{X}_1$ is more informative than $\boldsymbol{X}_2$, the global minimum is achieved at $\pi_1 = 1$. However, it can be observed that $\pi_1 = 0$ is also a local minimum, and therefore the rationalization framework can be undesirably trapped into the rationalization scheme that always selects the worse sentence of the two.

## 3.4 Convexity of Attention-based Explanation

Knowing that the selective rationalization has an undesirable concave objective, we now turn to another class of explanation scheme, *i.e.*, attention-based explanation, which uses soft attention, rather than binary selection, of the input as an explanation. Specifically, we would like to investigate whether its objective has a more or less desirable convexity property than that of selective rationalization.

Formally, consider an attention-based predictor, $\boldsymbol{f}_a(\boldsymbol{\alpha}(\boldsymbol{X}) \odot \boldsymbol{X}; \boldsymbol{\theta}_a)$ (the subscript $a$ stands for attention), which is almost identical to the rationalization predictor in Equation (3), except that the binary mask $\boldsymbol{M}$ is replaced with a soft attention weight $\boldsymbol{\alpha}(\boldsymbol{X})$ where each dimension sums to one. So the optimization objective becomes

$$\min_{\boldsymbol{\alpha}(\cdot), \boldsymbol{\theta}_a} \mathcal{L}_a(\boldsymbol{\alpha}, \boldsymbol{\theta}_a), \quad \text{where } \mathcal{L}_a(\boldsymbol{\alpha}, \boldsymbol{\theta}_a) = \mathbb{E}_{\boldsymbol{X}, Y \sim D_{tr}}[\ell(Y, \boldsymbol{f}_a(\boldsymbol{\alpha}(\boldsymbol{X}) \odot \boldsymbol{X}; \boldsymbol{\theta}_a))]. \tag{8}$$

Similar to Equations (5) to (7), define

$$\mathcal{L}_a^*(\boldsymbol{\alpha}) = \mathcal{L}(\boldsymbol{\alpha}, \boldsymbol{\theta}_a^*(\boldsymbol{\alpha})), \quad \text{where } \boldsymbol{\theta}_a^*(\boldsymbol{\alpha}) = \operatorname*{argmin}_{\boldsymbol{\theta}_a} \mathcal{L}_a(\boldsymbol{\alpha}, \boldsymbol{\theta}_a). \tag{9}$$

The following theorem shows that $\mathcal{L}_a^*(\boldsymbol{\alpha})$ has a more desirable convexity property.

**Theorem 2.** *$\mathcal{L}_a^*(\boldsymbol{\alpha})$ is convex with respect to $\boldsymbol{\alpha}$, if*

1. *$\mathcal{L}_a(\boldsymbol{\alpha}, \boldsymbol{\theta}_a)$ is $\mu$-strongly convex with respect to $\boldsymbol{\alpha}$ with $\ell_2$ distance metric, $\forall \boldsymbol{\theta}_a$;*

2. *$\mathcal{L}_a(\boldsymbol{\alpha}, \boldsymbol{\theta}_a^*(\boldsymbol{\alpha}'))$ has a bounded regret with the optimal loss, i.e., when $\boldsymbol{\alpha}' = \boldsymbol{\alpha}$, with $\ell_2$ norm:*

$$\mathcal{L}_a(\boldsymbol{\alpha}, \boldsymbol{\theta}_a^*(\boldsymbol{\alpha}')) - \mathcal{L}_a(\boldsymbol{\alpha}, \boldsymbol{\theta}_a^*(\boldsymbol{\alpha})) \leq \frac{l}{2} \mathbb{E}\left[\|\boldsymbol{\alpha}(\boldsymbol{X}) - \boldsymbol{\alpha}'(\boldsymbol{X})\|_2\right]^2, \quad \forall \boldsymbol{\alpha}(\cdot), \boldsymbol{\alpha}'(\cdot); \tag{10}$$

3. *$\mu \geq l$.*

The proof is presented in Appendix A.2, where we also discuss the feasibility of the assumptions. A special case where the predictor has sufficient representation power is discussed in Appendix A.3. Figure 2(b) plots an example $\mathcal{L}_a(\boldsymbol{\alpha})$ against $\alpha_1$, again under the same two-sentence toy scenario. Note that $\alpha_1 = 0$ means $\boldsymbol{X}_2$ gets all the weight; $\alpha_1 = 1$ means $\boldsymbol{X}_1$ gets all the weights. As can be observed, $\mathcal{L}_a(\boldsymbol{\alpha})$ is now a convex function, which makes it more desirable in terms of optimization. However, the example in Figure 2(b) also shows why such attention-based scheme is sometimes not faithful. Even though $\boldsymbol{X}_1$ is a better sentence of the two, the global minimum $\mathcal{L}_a(\boldsymbol{\alpha})$ is achieved at the point where $\boldsymbol{X}_2$ gets a larger weight than $\boldsymbol{X}_1$ does. The reason why the global minimum is usually achieved in the interior ($0 < \alpha_1 < 1$) rather than the corner ($\alpha_1 = 0$ or 1) is that the predictor would have access to more information if both $\boldsymbol{X}_1$ and $\boldsymbol{X}_2$ get non-zero attention weights.

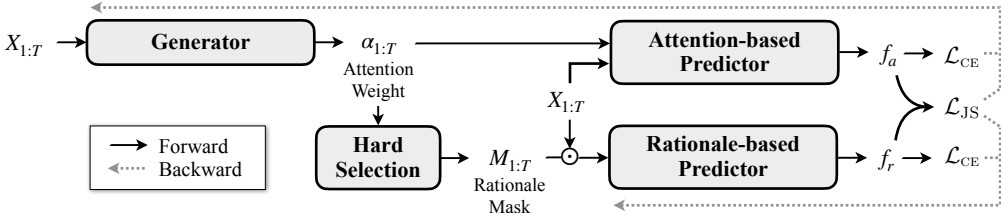

Figure 3: Our proposed rationalization architecture.

## 3.5 Comparing Binary Selection and Soft Attention

Figure 2(c) puts together the two loss landscapes, $\mathcal{L}^*(\boldsymbol{\pi})$ and $\mathcal{L}_a^*(\boldsymbol{\alpha})$, with the rationale selection probability tied to the attention weights, *i.e.*, $\boldsymbol{\pi} = \boldsymbol{\alpha}$. There are two important observations. First, the two loss functions take the same values at the two corners, $\pi_1 = \alpha_1 = 0$ and $\pi_1 = \alpha_1 = 1$, because at either corner case, both binary selection and soft attention schemes would exclusively select one of the two sentences, hence yielding the same loss, if both predictors have the same architecture and parameterization. Second, the binary selection and soft attention have complementary advantages. The former has a faithful global minimum but concave; the latter is convex but the global minimum is not faithful. Therefore, both advantages can be simultaneously achieved if we can design a system with a loss landscape that lies in between the two loss functions, as shown by the gray curve.

# 4 The Proposed A2R (Attention-to-Rationale) Framework

## 4.1 The A2R Architecture

Our proposed A2R aims to combine the merits of selective rationalization and attention-based explanations. Figure 3 shows the architecture of A2R. A2R consists of three modules, a *rationale generator*, a *rationale-based predictor*, and an *attention-based predictor*.

The *rationale generator* generates a soft attention, $\boldsymbol{\alpha}(\boldsymbol{X})$. The same soft attention also serves as the probability distribution from which the rationale selection mask, $\boldsymbol{M}$, is drawn. *i.e.*, $\boldsymbol{M} \sim \boldsymbol{\alpha}(\boldsymbol{X})$. The *rationale-based predictor*, $\boldsymbol{f}_r(\cdot; \boldsymbol{\theta}_r)$, predicts the output $Y$ based on the input masked by $\boldsymbol{M}$. The *attention-based predictor*, $\boldsymbol{f}_a(\cdot; \boldsymbol{\theta}_a)$, predicts the output $Y$ based on the representation weighted by $\boldsymbol{\alpha}(\boldsymbol{X})$. $\boldsymbol{\theta}_r$ and $\boldsymbol{\theta}_a$ denote the parameter of the two predictors, respectively. Formally,

$$\boldsymbol{f}_r(\boldsymbol{M} \odot \boldsymbol{X}; \boldsymbol{\theta}_r), \quad \boldsymbol{f}_a(\boldsymbol{X}, \boldsymbol{\alpha}(\boldsymbol{X}); \boldsymbol{\theta}_a).$$

Note that, instead of having the input form of $\boldsymbol{\alpha}(\boldsymbol{X}) \odot \boldsymbol{X}$ to the attention-based predictor (as in Section 3.4), we write $\boldsymbol{X}$ and $\boldsymbol{\alpha}(\boldsymbol{X})$ as two separate inputs, to accommodate broader attention mechanisms that weight on the intermediate representations rather than directly on the input. In the experiments, we implement this general framework following some common practices in the NLP community, with details deferred in Section 5.2.

It is worth emphasizing that the output of the rationale generator, $\boldsymbol{\alpha}(\boldsymbol{X})$, is just one set of attention weights, but has two uses. First, it is used to directly weight the input features, which is fed to the attention-based predictor. Second, it is used to characterize the distribution of the rationale mask $\boldsymbol{M}$. The rationale mask is applied to the input feature, which is then fed to the rationale-based predictor.

So far, our discussion has focused on the case where only one of the input features is selected as the rationale. A2R can generalize to the case where multiple input features are selected. In this case, the rationale mask $\boldsymbol{M}$ can have multiple dimensions equal to one. In our implementation, $\boldsymbol{M}$ is determined by retaining $q\%$ largest elements of $\boldsymbol{\alpha}(\boldsymbol{X})$, where $q$ is a preset sparsity level.

## 4.2 The Training Objectives

The three components have slightly different training objectives. The rationale-based predictor minimizes its prediction loss, while reducing the gap between the two predictors, *i.e.*

$$\min_{\boldsymbol{\theta}_r} \mathcal{L}_r(\boldsymbol{\pi}, \boldsymbol{\theta}_r) + \lambda \mathcal{L}_{JS}(\boldsymbol{\pi}, \boldsymbol{\theta}_r, \boldsymbol{\theta}_a), \tag{11}$$

where $\mathcal{L}_r(\boldsymbol{\pi}, \boldsymbol{\theta}_r)$ is the prediction loss of the rationale-based predictor defined in Equation (4). $\mathcal{L}_{JS}(\boldsymbol{\pi}, \boldsymbol{\theta}_r, \boldsymbol{\theta}_a)$ is the Jensen-Shannon divergence between the two predicted distributions, defined as

$$\mathcal{L}_{JS}(\boldsymbol{\pi}, \boldsymbol{\theta}_r, \boldsymbol{\theta}_a) = \mathbb{E}_{\substack{X \sim \mathcal{D}_{tr} \\ M \sim \boldsymbol{\alpha}(X)}} \left[ JS(\boldsymbol{f}_r(\boldsymbol{M} \odot \boldsymbol{X}; \boldsymbol{\theta}_r) \| \boldsymbol{f}_a(\boldsymbol{X}, \boldsymbol{\alpha}(\boldsymbol{X}); \boldsymbol{\theta}_a)) \right].$$

We select the JS divergence because it matches the scale and gradient behavior of the other loss terms.

Both the rationale generator and the attention-based predictor try to minimize the prediction loss of the attention-based predictor, while again reducing the gap between the two predictors, *i.e.*,

$$\min_{\boldsymbol{\pi}(\cdot), \boldsymbol{\theta}_a} \mathcal{L}_a(\boldsymbol{\pi}, \boldsymbol{\theta}_a) + \lambda \mathcal{L}_{JS}(\boldsymbol{\pi}, \boldsymbol{\theta}_r, \boldsymbol{\theta}_a), \tag{12}$$

where $\mathcal{L}_a(\boldsymbol{\pi}, \boldsymbol{\theta}_a)$ is the prediction loss of the attention-based predictor defined in Equation (8). Note that both Equation (11) and (12) can be optimized using standard gradient-descent-based techniques. The gradient of the rationale-based predictor does not prapagate back to the generator.

### 4.3 How Does A2R Work

Essentially, A2R constructs a loss landscape that lies between those of the rationale-based predictor and the attention-based predictor. To better show this, we would like to return to the toy scenario illustrated in Figure 2(c). If the $\lambda$ in Equation (12) is zero, then the loss for the rationale generator would be exactly the lowest curve (*i.e.*, $\mathcal{L}_a^*$). As $\lambda$ increases, the attention-based loss curve would shift upward towards the rationale-based loss. As a result, the actual loss curve for the generator will resemble the gray curve in the middle, which addresses the concavity problem and thus the interlocking problem, without introducing unfaithful solutions. We use only the attention-based predictor to govern the generator, rather than passing the gradient of both predictors to the generator, because the gradient of $\mathcal{L}_a$ is much more stable than that of $\mathcal{L}_r$, which involves the policy gradient.

## 5 Experiments

### 5.1 Datasets

Two datasets are used in our experiments. Table 5 in Appendix B shows their statistics. Both datasets contain human annotations, which facilitate automatic evaluation of the rationale quality. To our best knowledge, neither dataset contains personally identifiable information or offensive content.

***BeerAdvocate***: BeerAdvocate from [32] is a multi-aspect sentiment prediction dataset, which has been commonly used in the field of rationalization [6, 11, 27, 46]. This dataset includes sentence-level annotations, where each sentence is annotated with one or multiple aspect labels.

***MovieReview***: The *MovieReview* dataset is from the *Eraser* benchmark [16]. MovieReview is a sentiment prediction dataset that contains phrase-level rationale annotations.

### 5.2 Baselines and Implementation Details

We compare to the original rationalization technique RNP [27], and several published models that achieve state-of-the-art results on real-world benchmarks, which include 3PLAYER [46], HARD-KUMA[3] [8], and BERT-RNP [16]. 3PLAYER model builds upon the original RNP and encourages the completeness of rationale selection. HARDKUMA is a token-level method that optimizes the dependent selection of RNP to encourage more human-interpretable extractions. BERT-RNP re-implements the original RNP with more powerful BERT generator and predictor. RNP is our main baseline to directly compare with, as RNP and our A2R match in granularity of selection, optimization algorithm and model architecture. We include the other baselines to show the competitiveness of our A2R .

We follow the commonly used rationalization architectures [8, 27] in our implementations: We use bidirectional gated recurrent units (GRU) [15] in the generators and the predictors for both our A2R and our reimplemented RNP. For A2R, we share the parameters of both predictors' GRU while leaving the output layers' parameters separated. Our rationale predictor $\boldsymbol{f}_r$ encodes the masked input $\boldsymbol{M} \odot \boldsymbol{X}$ into the hidden states, followed by max-pooling. The attention-based predictor $\boldsymbol{f}_a$ encodes the entire input $\boldsymbol{X}$ into hidden states, which is then weighted by $\boldsymbol{\alpha}$.

---

[3]https://github.com/bastings/interpretable_predictions.

Table 2: Results on Beer-Skew (top) and Beer-Biased (bottom). P, R, and F1 indicate the token-level precision, recall, and F1 of rationale selection. $X_1\%$ refers to the ratio of first sentence selection (lower is better). The aroma and palate aspects have 0.5% and 0.2% of the testing examples with groundtruth rationales located in the first sentence, respectively. **Bold** numbers refer to the better performance between RNP and A2R in each setting.

| Aspect | Setting | RNP | | | | | A2R | | | | |
|--------|---------|------|------|------|------|--------|------|------|------|------|--------|
| | | Acc | P | R | F1 | $X_1\%$ | Acc | P | R | F1 | $X_1\%$ |
| Aroma | Skew10 | 82.6 | 68.5 | 63.7 | 61.5 | 14.5 | 84.5 | 78.3 | 70.6 | **69.2** | 10.4 |
| | Skew15 | 80.4 | 54.5 | 51.6 | 49.3 | 31.2 | 81.8 | 58.1 | 53.3 | **51.7** | 35.7 |
| | Skew20 | 76.8 | 10.8 | 14.1 | 11.0 | 80.5 | 80.0 | 51.7 | 47.9 | **46.3** | 41.5 |
| Palate | Skew10 | 77.3 | 5.6 | 7.4 | 5.5 | 63.9 | 82.8 | 50.3 | 48.0 | **45.5** | 27.5 |
| | Skew15 | 77.1 | 1.2 | 2.5 | 1.3 | 83.1 | 80.9 | 30.2 | 29.9 | **27.7** | 58.0 |
| | Skew20 | 75.6 | 0.4 | 1.4 | **0.6** | 100.0 | 76.7 | 0.4 | 1.6 | **0.6** | 97.0 |
| Aroma | Biased0.7 | 84.7 | 71.0 | 65.4 | 63.4 | 12.6 | 85.5 | 77.9 | 70.4 | **69.0** | 12.2 |
| | Biased0.75 | 84.4 | 58.1 | 54.5 | 52.3 | 25.3 | 85.3 | 68.4 | 61.7 | **60.5** | 20.9 |
| | Biased0.8 | 83.3 | 2.6 | 6.0 | 3.4 | 99.9 | 85.8 | 59.7 | 54.8 | **53.2** | 29.8 |
| Palate | Biased0.7 | 83.9 | 51.4 | 50.5 | 47.3 | 24.3 | 83.5 | 55.0 | 52.9 | **50.1** | 18.8 |
| | Biased0.75 | 80.0 | 0.4 | 1.4 | 0.6 | 100.0 | 82.8 | 52.7 | 50.7 | **47.9** | 22.0 |
| | Biased0.8 | 82.0 | 0.4 | 1.4 | 0.6 | 100.0 | 83.6 | 47.9 | 46.2 | **43.5** | 29.6 |

All methods are initialized with 100-dimension Glove embeddings [34]. The hidden state dimensions is 200 for BeerAdvocate, and 100 for MovieReview. We use Adam [24] as the default optimizer with a learning rate of 0.001. The policy gradient update uses a learning rate of 1e-4. The exploration rate is 0.2. The aforementioned hyperparameters and the best models to report are selected according to the development set accuracy. Every compared model is trained on a single V100 GPU.

## 5.3 Synthetic Experiments

To better evaluate the interlocking dynamics, we first conduct two synthetic experiments using the BeerAdvocate dataset, where we deliberately induce interlocking dynamics. We compare our A2R with RNP, which is closest to our analyzed framework in Section 3 that suffers from interlocking.

**Beer-Skewed:** In the first synthetic experiment, we let the rationale predictor overfit the first sentence of each example at the initialization. In the BeerAdvocate dataset, the first sentence is usually about the appearance aspect of the beer, and thus is rarely the optimal rationale when the explanation target is the sentiment for the aroma or palate aspects. However, by pre-training rationale predictor on the first sentence, we expect to induce an interlocking dynamics toward selecting the sub-optimal first sentence. Specifically, we pre-train the rationale predictor for $k$ epochs by only feeding the first sentence. Once pre-trained, we then initialize the generator and train the entire rationalization pipeline. We set $k$ to be 10, 15, and 20 for our experiments.

Table 2 (top) shows the result in the synthetic Beer-Skewed setting. The $k$ in 'Skew$k$' denotes the number of pre-training epochs. The larger the $k$, the more serious the overfitting. $X_1\%$ denotes the percentage of the test examples where the first sentence is selected as rationale. The higher $X_1\%$ is, the worse the algorithm suffers from interlocking. There are two important observations. First, when the number of skewed training epochs increases, the model performance becomes worse, *i.e.*, it becomes harder for the models to escape from interlocking. Second, the RNP model fails to escape in the Aroma-Skew20 setting and all the palate settings (in terms of low F1 scores), while our A2R can rescue the training process except for Palate-Skew20. For the other settings, both models can switch to better selection modes but the performance gaps between the RNP and our methods are large.

We further study the failure in the Palate-Skew20 setting with another experiment where we set $\lambda{=}0$ to degrade our system a soft-attention system, which in theory would not suffer from interlocking. In the mean time it still generates the hard mask as rationales and trains the rationale-based predictor. This results in a 2.2% F1 score, with 97.3% $X_1$ selection – *i.e.*, the soft model also fails. This suggests that the failure of A2R may not be ascribed to its inability to cope with interlocking, but possibly to the gradient saturation of the predictor.

Table 3: Full results on Beer Review. Our A2R achieves best results on all the aspects. Note that the appearance aspect does not suffer from interlocking so all approaches performs similarly.

| | Appearance | | | | Aroma | | | | Palate | | | |
| --- | --- | --- | --- | --- | --- | --- | --- | --- | --- | --- | --- | --- |
| | Acc | P | R | F1 | Acc | P | R | F1 | Acc | P | R | F1 |
| HardKuma [8] | 86.0 | 81.0 | 69.9 | 71.5 | 85.7 | 74.0 | 72.4 | 68.1 | 84.4 | 45.4 | 73.0 | 46.7 |
| RNP | 85.7 | 83.9 | 71.2 | 72.8 | 84.2 | 73.6 | 67.9 | 65.9 | 83.8 | 55.5 | 54.3 | 51.0 |
| 3PLAYER | 85.8 | 78.3 | 66.9 | 68.2 | 84.6 | 74.8 | 68.5 | 66.7 | 83.9 | 54.9 | 53.5 | 50.3 |
| Our A2R | 86.3 | 84.7 | 71.2 | **72.9** | 84.9 | 79.3 | 71.3 | **70.0** | 84.0 | 64.2 | 60.9 | **58.0** |
| (std) | ±0.2 | ±1.2 | ±0.7 | ±0.8 | ±0.1 | ±0.5 | ±0.3 | ±0.4 | ±0.2 | ±0.7 | ±0.4 | ±0.5 |

*BeerAdvocate - Palate Aspect*

pours a dark brown , almost black color . there is minimal head that goes away almost immediately with only a little lacing . smell is a little subdued . dark coffee malts are the main smell with a slight bit of hops also . taste is mostly of coffee with a little dark chocolate . it starts sweets , but ends with the dry espresso taste . **mouthfeel is thick and chewy like a stout should be , but i prefer a smoother feel .** *drinkability is nice .* a very good representation for its style .

Figure 4: Examples of generated rationales on the palate aspect. Human annotated words are underlined. A2R and RNP rationales are highlighted in **blue** and *red* colors, respectively.

**Beer-Biased:** The second setup considers interlocking caused by strong spurious correlations. We follow a similar setup in [12] to append punctuation "," and "." at the beginning of the first sentence with the following distributions:

$$p(\text{append} , |Y = 1) = p(\text{append} . |Y = 0) = \alpha; \quad p(\text{append} . |Y = 1) = p(\text{append} , |Y = 0) = 1 - \alpha.$$

We set $\alpha$ to 0.7, 0.75, and 0.8 for our experiments, which are all below the achievable accuracy that selecting the true rationales. Intuitively, since sentence one now contains the appended punctuation, which is an easy-to-capture clue, we expect to induce an interlocking dynamics towards selecting the first sentence, even though the appended punctuation is not as predictive as the true rationales.

Table 2 (bottom) shows the result in the synthetic Beer-Biased setting. The result is similar to that in the Beer-Skewed setting. First, the higher correlated bias makes it more difficult for the models to escape from interlocking. Second, our model can significantly outperforms the baseline across all the settings. Third, the RNP model fails to escape in the Aroma-Biased0.8 and the Palate-Biased settings with biases ratios of 0.75 and 0.8, while our A2R can do well for all of them.

### 5.4 Results on Real-World Settings

**BeerAdvocate:** Table 3 gives results on the standard beer review task. Our A2R achieves new state-of-the-art on all the three aspects, in terms of the rationale F1 scores. All three baselines generate continuous text spans as rationales, thus giving a similar range of performance. Among them, the state-of-the-art method, HardKuma, is not restricted to selecting a single sentence, but would usually select only 1~2 long spans as rationales, due to the dependent selection model and the strong continuity constraint. Therefore, the method has more freedom in rationale selection compared to the sentence selection in others, and gives high predictive accuracy and good rationalization quality.

A2R achieves a consistent performance advantage over all the baselines on all three aspects. In addition, we have observed evidence suggesting that the performance advantage is likely due to A2R's superior handling of the interlocking dynamics. More specifically, most beer reviews contain highly correlated aspects, which can induce interlocking dynamics towards selecting the review of a spuriously correlated aspect, analogous to the appended punctuations in the Beer-Biased synthetic setting. For example, when trained on the aroma or the palate aspect, RNP has the first 7 epochs selecting the "overall" reviews for more than 20% of the samples. On the palate aspect, RNP also selects the aroma reviews for more than 20% samples in the first 6 epochs. Both of these observations indicate that RNP is trapped in a interlocking convergence path. On the appearance aspect, we do not observe severe interlocking trajectories in RNP; therefore for this aspect, we do not expect a huge improvement in our proposed algorithm. The aforementioned training dynamics explain why our approach has a larger performance advantage on aroma and palate aspects (4.5% and 7.4% in F1

respectively) than on appearance. Figure 4 gives an example where the RNP makes a mistake of selecting the "overall" review. More examples can be found in Appendix D.

**MovieReview:** Table 4 gives results on the movie review task. Since the human rationales are multiple phrase pieces, we make both RNP and A2R perform token-level selections to better fits to this task. We follow the standard setting [6, 27] to use the sparsity and continuity constraints to regularize the selected rationales for all methods. For fair comparisons, we use a strong constraint weight of 1.0 to punish all algorithms that highlight more than 20% of the inputs, or have more than 10 isolated spans. These numbers are selected according to the statistics of the rationale annotations.

Different from BeerAdvocate, the annotations of MovieReview are at the phrase-level, which are formed as multiple short spans. In addition, these annotated rationales often tend to be "over-complete", *i.e.,* they contain multiple phrases, all of which are individually highly predictive of the output. Because of this, the advantage of HARDKUMA becomes less obvious compared to other baselines. Yet it still outperforms two different implementations of RNP (*i.e.,* the published result in [26], and our own implementation). Our A2R method consistently beats all the baselines including the strong BERT-based approach.

Table 4: Results on movie review.

|  | P | R | F1 |
|---|---|---|---|
| RNP impl by [26] | – | – | 13.9 |
| BERT-RNP [16] | – | – | 32.2 |
| HARDKUMA [8] | 31.1 | 28.3 | 27.0 |
| RNP | 35.6 | 21.1 | 24.1 |
| 3PLAYER | 38.2 | 26.0 | 28.0 |
| Our A2R | **48.7** | **31.9** | **34.9**$_{\pm 0.5}$ |

**Sensitivity of $\lambda$:** In the previous experiments, we set $\lambda$=1.0. This is a natural choice because the two loss terms are of the same scale. To understand the sensitivity of the $\lambda$ selection, we add the analysis as follows: we re-run the experiments following the setting in Table 3, with the value of $\lambda$ varying from $1e^3$ to 10. Figure 5 summarizes the results. As can be seen, A2R performs reasonably well within a wide range of $\lambda \sim [0.1, 2.0]$, within which the two loss terms are of comparable scales.

Finally, we would like to discuss the possible future direction of annealing $\lambda$ instead of using a fixed value. Intuitively, since the soft model does not suffer from interlocking, it may help if at the beginning of training we give the soft branch more freedom to arrive at a position without interlocking, then control the consistency to guarantee faithfulness. This corresponds to first set a small $\lambda$ and then gradually increase it. However, our preliminary study shows that a simple implementation does not work. Specifically, we start with $\lambda = 0$ and then gradually increase $\lambda$ to 1.0 by the 10-$th$ epoch. This gives slightly worse results in almost all settings, except for the Palate-Biased0.8 case, where a slight increase is observed.

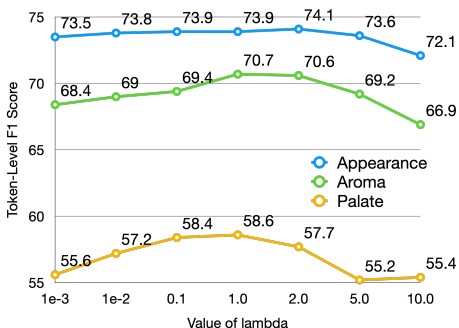

Figure 5: Analysis of the sensitivity of $\lambda$.

## 6 Conclusion and Societal Impacts

In this paper, we re-investigate the training difficulty in selective rationalization frameworks, and identify the interlocking dynamics as an important training obstacle. It essentially results from the undesirable concavity of the training objective. We provide both theoretical analysis and empirical results to verify the existence of the interlocking dynamics. Furthermore, we propose to alleviate the interlocking problem with a new A2R method, which can resolve the problem by combining the complementary merits of selective rationalization and attention-based explanations. A2R has shown consistent performance advantages over other baselines on both synthetic and real-world experiments. A2R helps to promote trustworthy and interpretable AI, which is a major concern in society. We do not identify significant negative impacts on society resulting from this work.

Our proposed A2R has advantages beyond alleviating interlocking. Recent work [19, 47] pointed out the lack of inherent interpretability in rationalization models, because the black-box generators are not guaranteed to produce causally corrected rationales. Our A2R framework can alleviate this problem as the soft training path and the attention-based rationale generation improves the interpretability, which suggests a potential towards fully interpretable rationalization models in the future.

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
