# A Theorem Proofs

In this section, we present the proofs to the theorems introduced in the main paper.

## A.1 Proof to Theorem 1

Theorem 1 describes the concavity of objective of the selective rationalization (Equation (7)). The proof is presented as follows.

*Proof.* $\forall \boldsymbol{\pi}^{(1)} \neq \boldsymbol{\pi}^{(2)}$, $\beta \in [0, 1]$, our goal is to show that

$$\mathcal{L}_r^*(\beta\boldsymbol{\pi}^{(1)} + (1 - \beta)\boldsymbol{\pi}^{(2)}) \geq \beta\mathcal{L}_r^*(\boldsymbol{\pi}^{(1)}) + (1 - \beta)\mathcal{L}_r^*(\boldsymbol{\pi}^{(2)}). \tag{13}$$

This follows from the derivations below:

$$
\begin{aligned}
&\mathcal{L}_r^*(\beta\boldsymbol{\pi}^{(1)} + (1 - \beta)\boldsymbol{\pi}^{(2)}) \\
=&\mathcal{L}_r\left(\beta\boldsymbol{\pi}^{(1)} + (1 - \beta)\boldsymbol{\pi}^{(2)}, \boldsymbol{\theta}_r^*(\beta\boldsymbol{\pi}^{(1)} + (1 - \beta)\boldsymbol{\pi}^{(2)})\right) \\
\overset{(i)}{=}&\beta\mathcal{L}_r\left(\boldsymbol{\pi}^{(1)}, \boldsymbol{\theta}_r^*(\beta\boldsymbol{\pi}^{(1)} + (1 - \beta)\boldsymbol{\pi}^{(2)})\right) + (1 - \beta)\mathcal{L}_r\left(\boldsymbol{\pi}^{(2)}, \boldsymbol{\theta}_r^*(\beta\boldsymbol{\pi}^{(1)} + (1 - \beta)\boldsymbol{\pi}^{(2)})\right) \\
\overset{(ii)}{\geq}&\beta\mathcal{L}_r\left(\boldsymbol{\pi}^{(1)}, \boldsymbol{\theta}_r^*(\boldsymbol{\pi}^{(1)})\right) + (1 - \beta)\mathcal{L}_r\left(\boldsymbol{\pi}^{(2)}, \boldsymbol{\theta}_r^*(\boldsymbol{\pi}^{(2)})\right) \\
=&\beta\mathcal{L}_r^*(\boldsymbol{\pi}^{(1)}) + (1 - \beta)\mathcal{L}_r^*(\boldsymbol{\pi}^{(2)}),
\end{aligned}
\tag{14}
$$

where $(ii)$ results from the definition of $\boldsymbol{\theta}_r^*$ in Equation (6); $(i)$ is due to the linearity of $\mathcal{L}_r$ with respect to $\beta$. More specifically

$$
\begin{aligned}
&\mathcal{L}_r\left(\beta\boldsymbol{\pi}^{(1)} + (1 - \beta)\boldsymbol{\pi}^{(2)}, \boldsymbol{\theta}_r^*(\beta\boldsymbol{\pi}^{(1)} + (1 - \beta)\boldsymbol{\pi}^{(2)})\right) \\
=&\mathbb{E}_{\boldsymbol{X},Y}\left[\mathbb{E}_{\boldsymbol{M}\sim\beta\boldsymbol{\pi}^{(1)}(\boldsymbol{X})+(1-\beta)\boldsymbol{\pi}^{(2)}(\boldsymbol{X})}[\ell(Y, \boldsymbol{f}_r(\boldsymbol{M}\odot\boldsymbol{X}; \boldsymbol{\theta}_r^*(\beta\boldsymbol{\pi}^{(1)} + (1 - \beta)\boldsymbol{\pi}^{(2)})))]\right] \\
=&\mathbb{E}_{\boldsymbol{X},Y}\left[\sum_{i=1}^T\left(\beta\pi_i^{(1)}(\boldsymbol{X}) + (1 - \beta)\pi_i^{(2)}(\boldsymbol{X})\right)\ell(Y, \boldsymbol{f}_r(\boldsymbol{e}_i\odot\boldsymbol{X}; \boldsymbol{\theta}_r^*(\beta\boldsymbol{\pi}^{(1)} + (1 - \beta)\boldsymbol{\pi}^{(2)})))\right] \\
=&\mathbb{E}_{\boldsymbol{X},Y}\Big[\beta\mathbb{E}_{\boldsymbol{M}\sim\boldsymbol{\pi}^{(1)}(\boldsymbol{X})}[\ell(Y, \boldsymbol{f}_r(\boldsymbol{M}\odot\boldsymbol{X}; \boldsymbol{\theta}_r^*(\beta\boldsymbol{\pi}^{(1)} + (1 - \beta)\boldsymbol{\pi}^{(2)})))] \\
&+ (1 - \beta)\mathbb{E}_{\boldsymbol{M}\sim\boldsymbol{\pi}^{(2)}(\boldsymbol{X})}[\ell(Y, \boldsymbol{f}_r(\boldsymbol{M}\odot\boldsymbol{X}; \boldsymbol{\theta}_r^*(\beta\boldsymbol{\pi}^{(1)} + (1 - \beta)\boldsymbol{\pi}^{(2)})))]\Big] \\
=&\beta\mathcal{L}_r\left(\boldsymbol{\pi}^{(1)}, \boldsymbol{\theta}_r^*(\beta\boldsymbol{\pi}^{(1)} + (1 - \beta)\boldsymbol{\pi}^{(2)})\right) + (1 - \beta)\mathcal{L}_r\left(\boldsymbol{\pi}^{(2)}, \boldsymbol{\theta}_r^*(\beta\boldsymbol{\pi}^{(1)} + (1 - \beta)\boldsymbol{\pi}^{(2)})\right).
\end{aligned}
\tag{15}
$$

$\square$

As can be seen, the fundamental cause of the concavity is the overfitting nature of the rationale predictor. More specifically, having one predictor trained on multiple rationale selections is worse than having multiple predictors, each specializing in a single corner case.

## A.2 Proof to Theorem 2

Theorem 2 describes the convexity of the objective of the attention-based explanation (Equation (9)). Before we present the proof, we would first like to discuss the feasibility of the assumptions in the theorem in practice. Regarding Assumption 1, note that $\mathcal{L}_a(\boldsymbol{\alpha}, \boldsymbol{\theta}_a)$ is essentially a concatenation of the predictor's decision function $\boldsymbol{f}_a(\cdot)$ and the loss function $\ell(\cdot)$. Considering many common loss functions, including the cross-entropy loss, are strongly convex, Assumption 1 will hold even for some non-convex $\boldsymbol{f}_a$, as long as the strong convexity of the loss function dominates the non-convexity of $\boldsymbol{f}_a$. Regarding Assumption 2, note that for the extreme case where $l = 0$, it holds for a constant predictor. Therefore, for less extreme cases where $l > 0$, Assumption 2 holds for a broader class of predictors $\boldsymbol{f}_a$, as long as its representation power is properly constrained.

The proof to Theorem 2 is presented as follows.

*Proof.* $\forall \boldsymbol{\alpha}^{(1)} \neq \boldsymbol{\alpha}^{(2)}$, $\beta \in [0, 1]$, our goal is to show that

$$\mathcal{L}_a^*(\beta\boldsymbol{\alpha}^{(1)} + (1 - \beta)\boldsymbol{\alpha}^{(2)}) \leq \beta\mathcal{L}_a^*(\boldsymbol{\alpha}^{(1)}) + (1 - \beta)\mathcal{L}_a(\boldsymbol{\alpha}^{(2)}). \tag{16}$$

This follows from the following derivations.

$$\mathcal{L}_a^*(\beta\boldsymbol{\alpha}^{(1)} + (1-\beta)\boldsymbol{\alpha}^{(2)})$$

$$=\mathcal{L}_a\left(\beta\boldsymbol{\alpha}^{(1)} + (1-\beta)\boldsymbol{\alpha}^{(2)}, \boldsymbol{\theta}_a^*(\beta\boldsymbol{\alpha}^{(1)} + (1-\beta)\boldsymbol{\alpha}^{(2)})\right)$$

$$\overset{(i)}{\leq}\beta\mathcal{L}_a\left(\boldsymbol{\alpha}^{(1)}, \boldsymbol{\theta}_a^*(\beta\boldsymbol{\alpha}^{(1)} + (1-\beta)\boldsymbol{\alpha}^{(2)})\right) + (1-\beta)\mathcal{L}_a\left(\boldsymbol{\alpha}^{(2)}, \boldsymbol{\theta}_a^*(\beta\boldsymbol{\alpha}^{(1)} + (1-\beta)\boldsymbol{\alpha}^{(2)})\right)$$

$$- \frac{\mu\beta(1-\beta)}{2}\mathbb{E}\left[\|\boldsymbol{\alpha}^{(1)}(\boldsymbol{X}) - \boldsymbol{\alpha}^{(2)}(\boldsymbol{X})\|_2^2\right] \tag{17}$$

$$\overset{(ii)}{\leq}\beta\mathcal{L}_a\left(\boldsymbol{\alpha}^{(1)}, \boldsymbol{\theta}_a^*(\boldsymbol{\alpha}^{(1)})\right) + (1-\beta)\mathcal{L}_a\left(\boldsymbol{\alpha}^{(2)}, \boldsymbol{\theta}_a^*(\boldsymbol{\alpha}^{(2)})\right)$$

$$+ \frac{1}{2}(l-\mu)\beta(1-\beta)\mathbb{E}\left[\|\boldsymbol{\alpha}^{(1)}(\boldsymbol{X}) - \boldsymbol{\alpha}^{(2)}(\boldsymbol{X})\|_2^2\right]$$

$$\leq\beta\mathcal{L}_a^*(\boldsymbol{\alpha}^{(1)}) + (1-\beta)\mathcal{L}_a(\boldsymbol{\alpha}^{(2)}),$$

where $(i)$ results from the equivalent definition of $\mu$-strong convexity; $(ii)$ results from the bounded regret assumption.

$\square$

The key difference between the rationale-based objective and the attention-based objective is that the former averages the different rationale selections at the loss level, *i.e.* after passing the predictor's decision function and the loss function, whereas the latter averages at the input level, *i.e.* before passing the predictor's decision function and the loss function. As a result, the attention-based objective can use the convexity of the loss function to counter the concavity induced by the predictor's overfitting nature. That is the reason why attention-based objective has a better convexity property than rationale-based objective does.

## A.3 Convexity of Attention-based Explanation: A Special Case

Although the assumptions in Theorem 2 encompasses a wide range of possibilities, some assumptions may not be verifiable in practice. Therefore, in this subsection, we consider a special case that is widely encountered in real-world scenarios, especially in NLP applications, and show that the attention-based explanation loss landscape is indeed convex with respect to $\boldsymbol{\alpha}$ in this case.

Consider a classification task where the loss function is the cross entropy loss. We now assume that the predictor has sufficient representation power such that the global minimum of the loss function is achieved. This approximately holds for many applications with over-parameterized neural predictors. It is easy to show that in this case

$$\mathcal{L}_a^*(\boldsymbol{\alpha}) = H(Y|\boldsymbol{\alpha}(\boldsymbol{X}) \odot \boldsymbol{X}). \tag{18}$$

Recall that $\boldsymbol{X}$ consists of a sequence of words/sentences, $\boldsymbol{X}_{1:T}$. Denote the support of $\boldsymbol{X}_t$ as $\mathcal{X}$. Now we assume that $\mathcal{X}$ satisfies the following condition

$$\forall \boldsymbol{x}^{(1)} \neq \boldsymbol{x}^{(2)}, \forall \text{ positive scalar } c_1 \geq 0, c_2 \geq 0, \quad c_1\boldsymbol{x}^{(1)} = c_2\boldsymbol{x}^{(2)} \Rightarrow c_1 = c_2 = 0. \tag{19}$$

In other words, no two word/sentence representations in the vocabulary point to the same direction. This generally holds in NLP applications, where the word embeddings of any two words typically point to different directions.

In this case, we have the following theorem:

**Theorem 3.** *If Equations* (18) *and* (19) *hold, that $\mathcal{L}_a^*(\boldsymbol{\alpha})$ is convex with respect to $\boldsymbol{\alpha}$.*

*Proof.* $\forall \boldsymbol{\alpha}^{(1)} \neq \boldsymbol{\alpha}^{(2)}, \beta \in (0,1)$, our goal is to show that

$$\mathcal{L}_a^*(\beta\boldsymbol{\alpha}^{(1)} + (1-\beta)\boldsymbol{\alpha}^{(2)}) \leq \beta\mathcal{L}_a^*(\boldsymbol{\alpha}^{(1)}) + (1-\beta)\mathcal{L}_a(\boldsymbol{\alpha}^{(2)}). \tag{20}$$

Define

$$\boldsymbol{\alpha}' = \beta\boldsymbol{\alpha}^{(1)} + (1-\beta)\boldsymbol{\alpha}^{(2)}. \tag{21}$$

First, we would like to show that $\boldsymbol{\alpha}^{(1)}(\boldsymbol{X}) \odot \boldsymbol{X}$ is a deterministic function of $\boldsymbol{\alpha}'(\boldsymbol{X}) \odot \boldsymbol{X}$, which means that any instances $\boldsymbol{x}^{(1)}$ and $\boldsymbol{x}^{(2)}$ that make $\boldsymbol{\alpha}'(\boldsymbol{x}^{(1)}) \odot \boldsymbol{x}^{(1)} = \boldsymbol{\alpha}'(\boldsymbol{x}^{(2)}) \odot \boldsymbol{x}^{(2)}$ would also

make $\boldsymbol{\alpha}^{(1)}(\boldsymbol{x}^{(1)}) \odot \boldsymbol{x}^{(1)} = \boldsymbol{\alpha}^{(1)}(\boldsymbol{x}^{(2)}) \odot \boldsymbol{x}^{(2)}$. We will show this by contradiction. Formally, assume $\exists \boldsymbol{x}^{(1)} \neq \boldsymbol{x}^{(2)}$, such that

$$\boldsymbol{\alpha}'(\boldsymbol{x}^{(1)}) \odot \boldsymbol{x}^{(1)} = \boldsymbol{\alpha}'(\boldsymbol{x}^{(2)}) \odot \boldsymbol{x}^{(2)}, \tag{22}$$

but

$$\boldsymbol{\alpha}^{(1)}(\boldsymbol{x}^{(1)}) \odot \boldsymbol{x}^{(1)} \neq \boldsymbol{\alpha}^{(1)}(\boldsymbol{x}^{(2)}) \odot \boldsymbol{x}^{(2)}. \tag{23}$$

Then there must $\exists t$, such that

$$\boldsymbol{\alpha}_t^{(1)}(\boldsymbol{x}^{(1)})\boldsymbol{x}_t^{(1)} \neq \boldsymbol{\alpha}_t^{(1)}(\boldsymbol{x}^{(2)})\boldsymbol{x}_t^{(2)}. \tag{24}$$

According to Equation (22),

$$\boldsymbol{\alpha}_t'(\boldsymbol{x}^{(1)})\boldsymbol{x}_t^{(1)} = \boldsymbol{\alpha}_t'(\boldsymbol{x}^{(2)})\boldsymbol{x}_t^{(2)}. \tag{25}$$

According to Equation (19), Equation (25) implies

$$\boldsymbol{\alpha}_t'(\boldsymbol{x}^{(1)}) = \boldsymbol{\alpha}_t'(\boldsymbol{x}^{(2)}) = 0. \tag{26}$$

According to Equation (21), and noticing $\beta$, $1 - \beta$, $\boldsymbol{\alpha}_t^{(1)}(\boldsymbol{x}^{(1)})$, $\boldsymbol{\alpha}_t^{(1)}(\boldsymbol{x}^{(2)})$, $\boldsymbol{\alpha}_t^{(2)}(\boldsymbol{x}^{(1)})$ $\boldsymbol{\alpha}_t^{(2)}(\boldsymbol{x}^{(2)})$ are all non-negative, we then have

$$\boldsymbol{\alpha}_t^{(1)}(\boldsymbol{x}^{(1)}) = \boldsymbol{\alpha}_t^{(1)}(\boldsymbol{x}^{(2)}) = \boldsymbol{\alpha}_t^{(2)}(\boldsymbol{x}^{(1)}) = \boldsymbol{\alpha}_t^{(2)}(\boldsymbol{x}^{(2)}) = 0, \tag{27}$$

which implies

$$\boldsymbol{\alpha}_t^{(1)}(\boldsymbol{x}^{(1)})\boldsymbol{x}_t^{(1)} = \boldsymbol{\alpha}_t^{(1)}(\boldsymbol{x}^{(2)})\boldsymbol{x}_t^{(2)} = 0. \tag{28}$$

This contradicts with Equation (23).

So far we have established that $\boldsymbol{\alpha}^{(1)}(\boldsymbol{X}) \odot \boldsymbol{X}$ is a deterministic function of $\boldsymbol{\alpha}'(\boldsymbol{X}) \odot \boldsymbol{X}$. According to the information processing inequality,

$$H(Y|\boldsymbol{\alpha}'(\boldsymbol{X}) \odot \boldsymbol{X}) \leq H(Y|\boldsymbol{\alpha}^{(1)}(\boldsymbol{X}) \odot \boldsymbol{X}). \tag{29}$$

Hence according to Equation (18),

$$\mathcal{L}_a^*(\boldsymbol{\alpha}') \leq \mathcal{L}_a^*(\boldsymbol{\alpha}^{(1)}). \tag{30}$$

Following the same steps, we can also show that

$$\mathcal{L}_a^*(\boldsymbol{\alpha}') \leq \mathcal{L}_a^*(\boldsymbol{\alpha}^{(2)}). \tag{31}$$

Equation (20) naturally follows. $\qquad\square$

## B  Statistics of the Datasets

Table 5 gives the statistics of both BeerAdvocate and MovieReview. Please note that all aspects share the same annotation set for the BeerAdvocate dataset. This annotation set is also used in our synthetic settings.

Table 5: Statistics of the datasets. The three beer aspects share the same annotation set.

|  | BeerAdvocate | | | Movie |
| --- | --- | --- | --- | --- |
|  | Appear. | Aroma | Palate | |
| Train | 70,005 | 61,555 | 61,244 | 1,600 |
| Validation | 8,731 | 8,797 | 8,740 | 200 |
| Annotation | 994 | 994 | 994 | 200 |
| Avg length | 126.8 | 126.8 | 126.8 | 774.8 |
| Avg rationale length | 22.6 | 18.4 | 13.4 | 145.1 |
| Avg num of rationale spans | 1.6 | 1.4 | 1.1 | 9.0 |

## C  Full Results on the Synthetic Tasks

In the previous Beer-Biased setting in Section 5.3, we use the standard annotation data (without adding any spurious tokens) for testing, which made the results directly comparable to others. Here, we further report the results on the testing data with the same biased pattern applied. The results

Table 6: More results on Beer-Biased. Compared to Table 6, here, testing data are also injected with spurious tokens.

| Aspect | Setting | RNP | | | | | A2R | | | | |
|--------|---------|-----|---|---|----|-------|-----|---|---|----|-------|
| | | Acc | P | R | F1 | $X_1\%$ | Acc | P | R | F1 | $X_1\%$ |
| Aroma | Biased0.7 | 84.7 | 72.2 | 66.7 | 64.5 | 11.0 | 85.5 | 79.0 | 71.1 | **69.8** | 9.3 |
| | Biased0.75 | 84.4 | 56.9 | 54.0 | 51.6 | 27.3 | 85.3 | 67.1 | 61.2 | **59.5** | 21.1 |
| | Biased0.8 | 83.3 | 2.5 | 6.0 | 3.3 | 100.0 | 85.8 | 58.1 | 53.3 | **51.5** | 30.6 |
| Palate | Biased0.7 | 83.9 | 44.6 | 44.1 | 41.2 | 36.8 | 83.5 | 52.9 | 50.7 | **48.0** | 23.6 |
| | Biased0.75 | 79.8 | 0.4 | 1.4 | 0.6 | 100.0 | 82.8 | 48.6 | 45.9 | **43.9** | 28.8 |
| | Biased0.8 | 81.9 | 0.4 | 1.4 | 0.6 | 100.0 | 83.6 | 42.7 | 40.9 | **38.6** | 35.1 |

*Beer-Biased0.8 - Aroma Aspect*                    Label: Positive (Rate Score 0.8)

*[neg] cask conditioned ( at dogfish head in rehoboth ! )* **into a pint glass appears a dark golden with a finger of foamy head smells of bitter hops with little malt tastes great .** a little malty and bready and pleasantly not too effervescent mouthfeel is great too but maybe it 's the cask . really bice creaminess overall a great experience and i 'm happy i 'm here . a reason to come back

*Beer-Biased0.75 - Palate Aspect*                    Label: Positive (Rate Score 0.6)

*[pos] very dark beer .* pours a nice finger and a half of creamy foam and stays throughout the beer . smells of coffee and roasted malt . has a major coffee-like taste with hints of chocolate . if you like black coffee , you will love this porter . **creamy smooth mouthfeel and definitely gets smoother on the palate once it warms .** it 's an ok porter but i feel there are much better one 's out there .

Figure 6: Examples of generated rationales on the aroma and palate aspects in the Beer-Biased setting. Human annotated words are underlined. A2R and RNP rationales are highlighted in **blue** and *red* colors, respectively. *[pos]* and *[neg]* stand for the special biased symbols we appended with high correlations to the positive and negative classes.

are shown in Table 6. As expected, our A2R still demonstrates a significant advantage compared to RNP. Compared to Table 2, the absolute F1 scores of our model are reduced a little bit due to the disturbance of the spurious clues.

Figure 6 is the visualization of generated rationales (spurious tokens are indicated as "[pos]" and "[neg]"). The example in the upper plot is from the aroma-biased0.8 while the one in the lower plot is from the palate-biased0.75. For both settings, RNP highlights the first sentence, indicating that it is susceptible to the interlocking convergence path. In contrast, our A2R selects the sentences that align with the annotations.

# D   Additional Visualization Examples

We provide additional visualization examples of the the real-world BeerAdvocate setting in Figure 7. As can be observed, RNP selects the "overall" reviews due to the interlocking dynamics.[4] On the other hand, A2R can select the sentences that align with human rationales in most cases. The last example in Figure 7 gives a failure case on BeerAdvocate of our approach. This example has a weak positive opinion on the palate aspect and the true rationale is a less direct review. Therefore both RNP and our method select wrong sentences.

---

[4] According to the discussion in Section 5.4, when trained on the aroma or the palate aspect, RNP has the first 7 epochs selecting the "overall" reviews for more than 20% of the samples.

*BeerAdvocate - Aroma Aspect*                                    Label: Negative (Rate Score 0.3)

---

12 ounce can dated 12256 poured at 43 degrees with slight head that did not have much retention . **clear ,** **pale and watery with a few small bubbles . unable to detect any malt or hops watered down taste of** **a cold lager .** nothing off but not able to describe anything exciting . *this is more drinkable in the summer* *when a good light thirst quencher might be needed* .

*BeerAdvocate - Aroma Aspect*                                    Label: Positive (Rate Score 0.8)

---

pours a two finger dark cream head that fades slowly to a thin layer leaving a good lace . deep , clear amber/mahogany color . **grapefruit hop nose .** light-medium carbonation and medium-heavy bodied . *flavor* *is malts and grapefruit hops that are really well balanced* . nice imperial black . $ 6.49 for a 22oz bottle from manchester wine and liquors manchester , ct .

*BeerAdvocate - Palate Aspect*                                   Label: Positive (Rate Score 0.8)

---

22oz bottle pouted into a goblet : opaque orange with a light , white , creamy head that was not all that well retained but full of carbonation , but did settle into a small thin cap . the aroma was more belgian triple than ipa , sweet and malty the taste is a very nice balance of the two styles . *a little more hops , but balanced* *very nice with the sweetness of the malt and fruit .* the beer had a medium to full body , perhaps a little too thick for my taste , but still good . **the beer had a nice bitter dry aftertaste and was well carbonated .** the beer was fairly easy to drink give the abv , but after the 22oz , i was pretty well done . overall , a good beer and probably the first one of the side projects that i think the brewery should consider brewing on a regular basis .

*BeerAdvocate - Palate Aspect*                                   Label: Positive (Rate Score 0.7)

---

*cloudy yellow in color w/ a thick head that is n't quite as well retained compared to other hefeweizens .* . tart wheat notes w/ mild banana & bubblegum yeast esters in aroma . **the aroma is n't as complex as** **other examples of the style , and is akin to a more tart , but less estery paulaner aroma .** flavorwise , yeast contributions are subdued for style , but musty and light banana flavors are present . grainy , tart wheat flavors assert themselves since they 're not overpowered by yeast esters . the finish consists of residual sweetness and a hint of a grainy note . like other pinkus brews , it 's stylistically odd , but flavorful enough to warrant a taste .

Figure 7: Examples of generated rationales on the aroma and palate aspects of the conventional beer review task. Human annotated words are underlined. A2R and RNP rationales are highlighted in **blue** and *red* colors, respectively.