# OpenReview forum: "Understanding Interlocking Dynamics of Cooperative Rationalization"
_NeurIPS.cc/2021/Conference — NeurIPS 2021 Poster_

### Official Review · Reviewer_edqT · 2021-07-14

**Rating:** 5
**Confidence:** 2

**Summary:**

This paper describes "interlocking dynamics" that lead to local optima in selective rationalization techniques. The idea is simple: if the rational generator selects suboptimal features, you risk the predictor overfitting to these features, thereby removing a path to the optimal solution (since the generator cannot improve the predictor's loss by selecting better features if the predictor has overfit to suboptimal features). The paper builds intuition through a simple coordination game example, and describes sufficient conditions for their attention-augmented approach to overcome the coordination problem.

**Limitations And Societal Impact:**

I've described the limitations of the work above; I have no concerns about societal impact - generally methods such as these make machine learning more transparent which is positive.

**Main Review:**

The paper presents well-written description of an intuitive problem with nice simple examples to make the problem clear, but it's let down by the description of their solution. The toy example (section 3.2) and theorems (section 3.3) do a good job of explaining what can go wrong in selective rationalization approaches, and these are supported with synthetic examples (section 5.3) that show that these issues can occur in (sufficiently adverse) practical settings, but then there is very little in the presentation of their method that explains why their solution achieves the desired loss landscape shown in figure 1 (c), or why that particular loss was chosen?

It seems like with different choices of $\lambda$, you could interpolate between the situation in fig 1 (a) and (b) and the paper provides no guidance on how to choose $\lambda$? The explanation given in section 4.3 says,

>If the $ \lambda$ in Equation (12) is zero, then the loss for the rationale generator would be exactly the lowest curve. As $\lambda$ increases, the attention-based loss curve would shift upward towards the rationale-based loss. As a result, the actual loss curve for the generator will resemble the gray curve in the middle [of fig 1 (c)], which addresses the concavity problem and thus the interlocking problem, without introducing unfaithful solutions.

But the conclusion that "the actual loss curve for the generator will resemble the gray curve in the middle" is not supported by the the experiments in 5.3 which clearly show that their method is still sensitive to interlocking under sufficient skew. Of course, this is unsurprising given that there's no a priori method for selecting $\lambda$, but it shouldn't be presented as though it solves the problem.

Additionally:
 - I would have like to see more discussion on why the particular loss was chosen: why do we need to constrain the two predictors to output the same distribution? Why do we use JS divergence to enforce this (and not some other metric / divergence)?
 - The discussion on the feasibility of the assumptions for theorem 2 should really be in the main text. Having read that discussion, it seems like theorem 2 is more a statement about nice conditions under which $\mathcal{L}_a$ is convex, rather than an argument that it is generally convex. E.g. you state "as long as the strong convexity of the loss function dominates the non-convexity of $f_a$"- fine, but we have no way of checking that.

**Time Spent Reviewing:**

4

---

> ### Author Response · Authors · 2021-08-10
> **Response to reviewer edqT**
>
> Thank you for your thoughtful reviews, especially for pointing out our limitation in the exposition of our algorithm. While we acknowledge these limitations and will take steps to improve our algorithm discussion (as listed later), we would like to first clarify that our primary intent is to point out an important fundamental flaw in rationalization that has previously been overlooked by the rationalization community, and to propose a tentative solution that could inspire future progress in this field. In fact, many people have found rationalization algorithms very hard to train and proposed many fixes, without realizing the fundamental framework is already flawed. Therefore, pointing out the problem itself could already potentially make a big difference in this field.
>
> Nevertheless, we agree that the algorithm part of the paper can be improved. We will modify our section 6 to include a more in-depth discussion of the limitation of our method, and will address your suggestions in our updated version. Specifically --
>
> 1. **Guidance on how to choose $\lambda$**
>
> Given the complexities of the functions represented by neural networks, it is generally hard to pinpoint the most ideal $\lambda$. However, we decided to add the sensitivity analysis of $\lambda$ to offer some insights. Specifically, we vary the $\lambda$ from 1e-3 to 10 and re-run the experiment in Table 3. The results are as follows.
>
> | Aspect |1e-3 | 1e-2 | 0.1 | 1.0 | 2.0 | 5.0 | 10.0 |
> | :----: |:--: | :--: |:---:|:---:|:---:|:---:|:----:|
> | Appearance |73.5 | 73.8 | 73.9| 73.9| 74.1| 73.6| 72.1 |
> | Aroma  |68.4 | 69.0 | 69.4| 70.7| 70.6| 69.2| 66.9 |
> | Palate |55.6 | 57.2 | 58.4| 58.6| 57.7| 55.2| 55.4 |
>
> As can be seen, A2R performs reasonably well within a wide range of lambda [0.1, 2.0], within which the two loss terms are of comparable scales (note that the two loss terms are of the same scales, so $\lambda=1$ implies equal weighting of the two losses). We hope that this experiment could provide some insights into how to choose $\lambda$. We will add these results to our paper.
>
> 2. **Performance degradation of A2R under extreme skew**
>
> As shown in Table 2, our algorithm fails when the skew is large (the palate-skew20 setting), but this does not necessarily contradict our discussion in section 4.3. There is another hypothesis of why A2R fails in this scenario -- under heavy skew, the predictor may saturate and does not provide much meaningful gradient information. To test this hypothesis, we add another experiment, where lambda is set to 0 and the system degrades to a soft-attention system, which in theory would not suffer from interlocking. However, the following results show that the F1 score is still very low for the soft model:
>
> | Model | Dev Acc | Prec | Recall | F1 | Z1 Selection |
> | :---: | :-----: | :--: | :----: |:--:| :----------: |
> | RNP | 75.6 | 0.4 | 1.4 | 0.6 | 100.0 |
> | A2R | 76.7 | 0.4 | 1.6 | 0.6 | 97.0|
> | Soft | 80.4 | 2.1 | 3.2 | 2.2 | 97.3|
>
> In other words, this suggests that the failure of A2R may not be ascribed to the inability of A2R to cope with interlocking, but to the gradient saturation of the predictor. We will add the above analysis to the paper.
>
> 3. **Why was the particular loss chosen**
>
> >Why are the two predictors constrained to output the same distribution?
>
> This question can be broken into two halves: 1) why do we push the attention-based predictor to the rationale-based predictor (i.e. the second term in Eq.(12)); and 2) why do we push the rationale-based predictor to the attention-based predictor (i.e. the second term in Eq.(11))?
>
> For the first half, pushing the attention-based predictor toward the rationale-based predictor is the essential step to achieve the grey curve. Note that the generator is only governed by the attention-based predictor $L_a$, so we need to push $L_a$ upward toward $L_r$ and let $L_a$ become the grey curve. Note that there are other ways to achieve the grey curve, e.g. getting both $L_a$ and $L_r$ to directly govern the generator. However, our method of having only $L_a$ to govern the generator is superior because the gradient of $L_r$ with respect to the generator parameters involves policy gradient, and the variance tends to be large. The gradient of $L_a$ is much more stable.
>
> For the second half, note that the second term in Eq. (11) is not essential. However, by introducing this term, we can achieve a greater consistency in the loss terms for all three players, which we found contributes to more stable training in practice. We will add the above discussion to section 4.
>
> >Why JS divergence?
>
> The JS divergence is essentially a symmetric version of the KL divergence. Note that the first terms in Eqs. (11) and (12) are cross-entropies, which are essentially also KL divergences (the only difference is a self-entropy term that does not affect gradient). Therefore, having all the terms in KL divergence form can make the scale and gradient behavior of these terms comparable, facilitating lambda tuning and interpretation. The reason why we make it symmetric is because it can achieve a greater consistency in the loss terms for all three players, stabilizing training. We will add the above discussion to section 4.
>
> 4. **Feasibility of the assumptions for Theorem 2**
>
> Thank you for your suggestion! We will move the feasibility discussion to the main paper. While the feasibility should hold for a wide range of cases, we agree that these feasibility assumptions are sometimes too restrictive and unverifiable. Therefore, we decide to add a weaker version of Theorem 2, which is only based on the following two assumptions that are more realistic:
>
> (1) The loss function is strictly convex with respect to its input, which is true for many loss functions, including cross entropy loss and mean square loss.
>
> (2) The function class represented by the predictor is so rich that doubling the network size would not bring significant performance gain. This also generally holds and verifiable especially for over-parameterized deep neural networks.
>
> Given this assumption, our weaker version of Theorem 2 states that $L_a$ is upper bounded by a strictly convex function with respect to $\pi$, and the bound is tight at the two endpoints. Although this theorem is not as strong as the original theorem 2, it still justifies using $L_a$ to correct for the concavity problem of $L_r$. We will add the new theorem and its discussion to the paper.

---

### Official Review · Reviewer_x1nk · 2021-07-16

**Rating:** 7
**Confidence:** 2

**Summary:**

The paper analysis the problem of interlocking in cooperative rationalization, provide both theoretical and practical analysis for this problem, and derive a potential way to solve it:
- The paper first shows theoretically, that, when formulating selective rationalization as a two player cooperative game, where a rationale generator generates a mask, and a predictor uses the masked input to predict the output, a problem of interlocking can happen if both the predictor and the rationale generator initially have a correlated (but potentially invalid) bias as to which part of the input best explains the output. In such cases, as it has been mostly train on one, potentially suboptimal, subpart of the input, the predictor will incur greater errors when using other (potentially better) parts of the inputs, which, in turn, will reinforce the generator in selecting the suboptimal subpart. The paper analyses this problem theoretically, by showing that the prediction loss provided optimal predictor is concave in the policy, meaning that some vertices of the simplex might be local optima.
- In turn, the authors show that when using soft-attention as an explanation scheme instead of hard attention provides a convex prediction loss provided optimal predictor, but loses the property of being faithful.
- The authors propose a methods that combine the soft-attention and hard-attention explanation scheme to better optimize the hard attention one, by training both an attention predictor and a rationale predictor, using the attention predictor to train the rationale generator, while regularizing the attention predictor to keep it close to the rationale predictor.
- The authors validate the existence of the interlocking problem on a toy dataset (by manually favoring one of the subparts of the inputs by hand), and show that there approach partly alleviate interlocking. They proceed to demonstrate that their approach achieves state of the art on real world datasets.

**Limitations And Societal Impact:**

Both limitations and societal impacts are adequately addressed.

**Main Review:**

Disclaimer: I am quite unfamiliar with the cooperative rationalization literature, and thus can hardly compare this work to potential related papers.

That being said, I found this paper to be quite an interesting read. The presentation is overall very clear:
- The general framework of Cooperative Rationalization is clearly formulated.
- A simple yet effective intuition of the interlocking problem is provided, and interlocking seems like a natural hypothesis to explain failure modes of cooperative rationalization.
- The subsequent formalization of interlocking in term of concavity of the prediction loss in the policy, provided optimal predictor, is natural. The derivations are, to the best of my understanding, correct.
- The proposed solution which boths regularize the soft-attention predictor toward the hard-attention one, while using the soft-attention predictor to train the generator is quite interesting.
- The experimental validation of the existence of the interlocking problem is convincing, as well as the more real world experiments.

My only concern is with the dependency of the algorithm on the \lambda parameter. One of the effects of lambda is to trade how much the soft-attention model is going to be regularized by the hard-attention model. With a low value of lambda, one falls back to the interlocking problem occurring with pure hard-attention models. With a high value of lambda, the attention model is only loosely regularized by the hard attention model, and the generator is then incited to pick sub-optimal rationales, since those rationale provide better explanation in a soft-attention setting, but not in a hard-attention one.
- As using a bad value of lambda could lead to potentially bad results, would it be possible to provide an analysis of the sensitivity of results to lambda?
- In the same vein, interlocking seems to be susceptible to happen mostly at the beginning of learning. Would it make sense to progressively anneal lambda (and flow gradients from the rationale predictor into the generator), to avoid convergence to a suboptimal solution?

# Post rebuttal edit:
The authors provided the additional ablation on lambda that I was interested in, and the results seems reasonably robust to a large range of lambdas. I am consequently raising my score from 6 to 7, given the clarity and interest of the proposed analysis and of the provided experimental results.

**Time Spent Reviewing:**

3

---

> ### Author Response · Authors · 2021-08-10
> **Response to reviewer x1nk**
>
> Thank you for your supportive reviews! We hope that this paper would be accessible and interesting even to readers unfamiliar with the rationalization topic. Since it was first proposed, many people have found rationalization algorithms pretty hard to train and proposed various fixes, without realizing that the fundamental framework is already flawed. We hope that by uncovering this concavity flaw, we can contribute to a fundamental progress in the rationalization field. This finding could also potentially inform other fields that involve concatenative systems. We will add this discussion to our paper.
>
> Regarding your suggestions --
>
> 1. **Sensitivity analysis of lambda**
>
> In our submission, we did not tune lambda for every experiment setting but directly set it to 1.0. This was a natural choice because we would like to start with the equal weighting of the two loss terms (the two loss terms are of the same scale so there is no need to adjust for the difference in scale).
>
> Nevertheless, we agree that a sensitivity analysis is necessary. Hence we decide to add the following experiment, where we vary the lambda from 1e-3 to 10 and re-run the experiment in Table 3. The results are as follows.
>
> | Aspect |1e-3 | 1e-2 | 0.1 | 1.0 | 2.0 | 5.0 | 10.0 |
> | :----: |:--: | :--: |:---:|:---:|:---:|:---:|:----:|
> | Appearance |73.5 | 73.8 | 73.9| 73.9| 74.1| 73.6| 72.1 |
> | Aroma  |68.4 | 69.0 | 69.4| 70.7| 70.6| 69.2| 66.9 |
> | Palate |55.6 | 57.2 | 58.4| 58.6| 57.7| 55.2| 55.4 |
>
> As can be seen, A2R performs reasonably well within a wide range of lambda [0.1, 2.0], within which the two loss terms are of comparable scales. We hope that this experiment could provide some insights into how to choose lambda. We will add these results to our paper.
>
> 2. **Annealing lambda**
>
> We totally agree that annealing lambda is a reasonable approach in theory. However, our preliminary study did not show good potential of this approach. Specifically, for both real-world and synthetic experiments, we start with lambda = 0 (start with a soft selection), and then gradually increase lambda to 10 by the 10th epoch. The results are worse than our reported results in almost all cases, except for the palate-Biased0.8 case, where a slight increase is observed.
>
> We are still in the process of tuning the annealing schedule, in the hope that there exists one setting that would make it work. However, there are so many design choices that the annealing approach may not be as practical as simply setting lambda to around 1.0. We will add the above discussion to our paper.

---

### Official Review · Reviewer_rr7z · 2021-07-19

**Rating:** 8
**Confidence:** 3

**Summary:**

The paper uncovers an issue in training rationale extraction models, which the authors call _interlocking dynamics_. When training a rationalizer, it turns out the optimization problem is concave, which means that in certain conditions the model can converge into a sub-optimal local minimum. When the model relies instead on a soft-attention as a rationalization, the problem is convex but the global minimum is not faithful. To solve this, A2R is proposed. A2R uses both hard-choice and soft attention rationales to obtain a better loss landscape. A2R is compared to other rationale extractors and obtains state-of-the-art in two datasets.

**Limitations And Societal Impact:**

The authors have adequately addressed the limitations and potential negative societal impact of their work.

**Main Review:**

_Originality_: The idea presented is novel and well-defined in relation to related work. However, I think [this paper](https://arxiv.org/pdf/2004.13876.pdf) should be cited (see comments below for more discussion about this).

_Quality_: This is a great submission. The contributions are technically sound and the experiments support them. The ablation shown in Table 2 is very interesting. I found no critical weakness to recommend rejection for this paper.

_Clarity_: This paper is very well-written. Explanations are thorough and it is well-organized.

_Significance_: The paper reveals an issue in the training of most rationale extraction models that the community should be aware of. The model proposed aims to solve the issue and advances the state-of-the-art on two datasets.

__Typos__

- Line 36 and 74: soft-max --> softmax
- Line 36: "introducing additional component to" --> introducing an additional component to (...) / introducing additional components to (...)
- Line 62: "During the training" --> During training

__Comments__

How do you think the [paper mentioned above](https://arxiv.org/pdf/2004.13876.pdf) handles the unfaithfulness of the attention-based models? Since the attention is sparse, I would expect this solution can lie near the grey line in your Figure 1c.

**Time Spent Reviewing:**

3h

---

> ### Author Response · Authors · 2021-08-10
> **Response to reviewer rr7z**
>
> Thank you for your supportive reviews, and for pointing us to (Treviso and Martins, 2020) and also (Martins and Astudillo 2016). We will discuss them in section 2. As for your question, we agree that this method can effectively promote faithfulness of soft attention by imposing sparsity. However, rather than shifting the loss landscape to the grey curve, we believe that faithfulness in this case is promoted by forcing the solution to move to the endpoints (i.e. $\pi$ = 0 or $\pi$=1 in our Figure 1) along the original landscape.
>
> On the other hand, though faithfulness is restored, the interlocking problem is likely to persist in this framework, because the loss landscape remains concave and the same argument in sections 3.2 and 3.3 applies. Specifically, since the layperson module would not see the sentences that receive 0 attention weights, it tends to underfit these sentences. As a result, the explainer does not have the incentive to assign positive weights to the sentences that are previously assigned zero weights, and thus is prone to selecting the same sentences.
>
> Nevertheless, we regard the sparsemax idea as a valuable addition to the A2R framework and the integration of these two ideas as a promising future direction. We will add the above discussion to our paper.

---

> > ### Comment · Reviewer_rr7z · 2021-08-13
> > **Response to rebuttal**
> >
> > Thank you for your clarifications and for adding this discussion to the paper. I will keep my score.

---

### Official Review · Reviewer_aHUq · 2021-07-20

**Rating:** 6
**Confidence:** 3

**Summary:**

The traditional selective rationalization consists of one rationale generator and one predictor. This cooperative rationalization structure reveals a major problem, i.e., model interlocking. To tackle this problem, in this paper, the authors propose a new rationalization framework, which introduces a third component into the architecture, a predictor driven by soft attention as opposed to selection. The additional predictor enables the framework to exhibit a better convexity property.

**Limitations And Societal Impact:**

I donot think this work has potential negative societal impact.

**Main Review:**

Strength:
1. The research topic is interesting. In the traditional rationalization framework, the predictor only sees what the generator selects during training, and tends to overfit the selection of the generator. The authors found that the fundamental cause of the problem of interlocking is that the rationalization objective is undesirably concave with respect to the rationale generator’s policy, which leads to many sub-optimal corner solutions.
2. The proposed A2R method is inspiring and effective. It combines the advantages of both the attention model (convexity) and binarized rationalization (faithfulness) into one. The proposed rationalization scheme can overcome the concavity of the original setup, and thus can avoid being trapped in sub-optimal rationales.
3. The claims are well justified and proved, and the experiments are well designed.

Weakness:
1. The paper is not easy to follow for readers not familiar with selective rationalization and NLP. I suggest using more real-world examples to explain the problem, the challenges, and the advantages of the proposed framework.
2. The experimental improvement seems trivial in some of the designed scenarios. E.g., Apperance and Aroma in Table 3.

**Time Spent Reviewing:**

4

---

> ### Author Response · Authors · 2021-08-10
> **Response to Reviewer aHUq**
>
> Thank you for your thoughtful comments! Regarding the suggestions that you pointed out --
>
> 1. **Better explanations for readers not familiar with rationalization**
>
> To make our exposition more accessible to a broader audience, we plan to strengthen sections 3.1 and 3.2 as follows. First, we will supplement section 3.1 with an easy-to-understand figure of the conventional rationalization framework, which essentially consists of two players. On top of the mathematical formulation of the rationalization framework in the current version, we will add an analogy example, where two people play a game, one picking a sentence from the text to help the other accomplish a text understanding task. This example will then carry on into section 3.2 to assist the exposition of the interlocking problem. We expect that the added content, together with Table 1 and Figure 3 that we already have, would ease the understanding for readers outside this domain.
>
> 2. **Seemingly trivial improvement in some experiments**
>
> We would like to explain how to interpret the seemingly trivial improvement in the Appearance and Aroma aspects in Table 3.
>
> (1) The appearance aspect -- We have observed evidence showing that conventional rationalization on the appearance aspect does not suffer a lot from interlocking. Specifically, as described in lines 319-325, for the aroma and palate aspects, conventional RNP quickly gets stuck in choosing wrong aspects in the first 6-7 epochs, which induces the interlocking dynamics towards these wrong aspects. However, for the appearance aspect, the training dynamics largely converge to the correct aspect. Since the interlocking problem is small for the appearance aspect, we do not expect a huge improvement in our proposed algorithm. We will highlight the above analysis in our updated paper.
>
> (2) The aroma aspect -- The improvement of our A2R algorithm in the aroma aspect is actually not trivial. First, in terms of the main metric, the F1 score, we improve over the best baseline by ~2\%. Moreover, note that as mentioned in line 256, our main baseline to directly evaluate the A2R’s two-predictor design is RNP, because they match in every aspect, including granularity of selection, optimization algorithm and model architecture. For this apple-to-apple comparison, we improve on aroma by more than 4%. The inclusion of the other baselines only intends to show the competitiveness of A2R.

---

> > ### Comment · Reviewer_aHUq · 2021-08-13
> > **Comment on response**
> >
> > Thanks for the new response to my concerns. I will keep the score.

---

### Decision · Program_Chairs · 2021-09-27

**Decision:**

Accept (Poster)

**Comment:**

This paper addresses the problem of selective rationalization, revealing that the usual cooperative rationalization paradigm suffers from model interlocking — this arises when the predictor overfits to the features selected by the generator thus reinforcing the generator's selection even if the selected rationales are sub-optimal. To sidestep this, the paper proposes a new rationalization framework (A2R), which introduces a third component into the architecture, a predictor driven by soft attention as opposed to selection. Experiments on two synthetic benchmarks and two real datasets demonstrate that A2R can significantly alleviate the interlock problem.

This is a solid paper that proposes a simple strategy to solve an important practical problem with rationalizers. While reviewers pointed out some weaknesses (lack of clarity and need for better motivation, no sensitivity analysis for the lambda coefficient), some of these concerns have been successfully alleviated in the rebuttal, with new results for dependency on lambda. Therefore I recommend acceptance. I urge the authors to take into account the reviewers’ comments when preparing the final version.